# Learning Hard Optimization Problems:
# A Data Generation Perspective

**James Kotary**
Syracuse University
Syracuse, NY 13244
jkotary@syr.edu

**Ferdinando Fioretto**
Syracuse University
Syracuse, NY 13244
ffiorett@syr.edu

**Pascal Van Hentenryck**
Georgia Institute of Technology
Atlanta, GA 30332
pvh@isye.gatech.edu

## Abstract

Optimization problems are ubiquitous in our societies and are present in almost every segment of the economy. Many of these optimization problems are NP-hard and computationally demanding, often requiring approximate solutions for large-scale instances. Machine learning frameworks that learn to approximate solutions to such hard optimization problems are a potentially promising avenue to address these difficulties, particularly when many closely related problem instances must be solved repeatedly. Supervised learning frameworks can train a model using the outputs of pre-solved instances. However, when the outputs are themselves approximations, when the optimization problem has symmetric solutions, and/or when the solver uses randomization, solutions to closely related instances may exhibit large differences and the learning task can become inherently more difficult. This paper demonstrates this critical challenge, connects the variation of the training data to the ability of a model to approximate it, and proposes a method for producing (exact or approximate) solutions to optimization problems that are more amenable to supervised learning tasks. The effectiveness of the method is tested on hard non-linear nonconvex and discrete combinatorial problems.

## 1 Introduction

Constrained optimization (CO) is in daily use in our society, with applications ranging from supply chains and logistics, to electricity grids, organ exchanges, marketing campaigns, and manufacturing to name only a few. Two classes of hard optimization problems of particular interest in many fields are (1) *combinatorial optimization problems* and (2) *nonlinear constrained problems*. Combinatorial optimization problems are characterized by discrete search spaces and have solutions that are combinatorial in nature, involving for instance, the selection of subsets or permutations, and the sequencing or scheduling of tasks. Nonlinear constrained problems may have continuous search spaces but are often characterized by highly nonlinear constraints, such as those arising in electrical power systems whose applications must capture physical laws such as Ohm's law and Kirchhoff's law in addition to engineering operational constraints. Such CO problems are often NP-Hard and may be computationally challenging in practice, especially for large-scale instances.

While the AI and Operational Research communities have contributed fundamental advances in optimization in the last decades, the complexity of some problems often prevents them from being adopted in contexts where many instances must be solved over a long-term horizon (e.g., multi-year planning studies) or when solutions must be produced under time constraints. Fortunately, in many practical cases, including the scheduling and energy problems motivating this work, one is interested in solving many problem instances sharing similar patterns. Therefore, the application of deep learning methods to aid the solving of computationally challenging constrained optimization problems appears to be a natural approach and has gained traction in the nascent area at the intersection

35th Conference on Neural Information Processing Systems (NeurIPS 2021).

between CO and ML [5, 19, 31]. In particular, supervised learning frameworks can train a model using pre-solved CO instances and their solutions. However, learning the underlying combinatorial structure of the problem or learning approximations of optimization problems with hard physical and engineering constraints may be an extremely difficult task. While much of the recent research at the intersection of CO and ML has focused on learning good CO approximations in jointly training prediction and optimization models [3, 18, 22, 25, 32] and incorporating optimization algorithms into differentiable systems [1, 27, 34, 20], learning the combinatorial structure of CO problems remains an elusive task.

Beside the difficulty of handling hard constraints, which will almost always exhibit some violations, two interesting challenges have emerged: the presence of multiple, often symmetric, solutions, and the learning of approximate solution methods. The first challenge recognizes that an optimization problem may not have a unique solution. This challenge is illustrated in Figure 1, where the various $\boldsymbol{y}^{(i)}$ represent *optimal* solutions to CO instances $\boldsymbol{x}^{(i)}$ and $\mathcal{C}$ the feasible space. As a result, a combinatorial number of possible datasets may be generated. While equally valid as optimal solutions, some sets follow patterns which are more meaningful and recognizable. Symmetry breaking is of course a major area of combinatorial optimization and may alleviate some of these issues. But different instances may not break symmetries in the same fashion, thus creating datasets that are harder to learn.

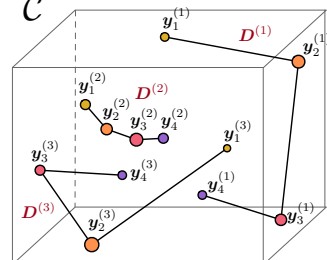

Figure 1: Co-optimal datasets due to symmetries.

The second challenge comes from realities in the application domain. Because of time constraints, the solution technique may return a sub-optimal solution. Moreover, modern combinatorial optimization techniques often use randomization and large neighborhood search to produce high-quality solutions quickly. Although these are widely successful, different runs for the same, or similar, instances may produce radically different solutions. As a result, learning the solutions returned by these approximations may be inherently more difficult. These effects may be viewed as a source of noise that obscures the relationships between training data and their target outputs. Although this does not raise issues for optimization systems, it creates challenging learning tasks.

This paper demonstrates these relations, connects the variation in the training data to the ability of a model to approximate it, and proposes a method for producing (exact or approximate) solutions to optimization problems that are more amenable to supervised learning tasks. More concretely, the paper makes the following contributions:

1. It shows that the existence of co-optimal or approximated solutions obtained by solving hard CO problems to construct training datasets challenges the learnability of the task.

2. To overcome this limitation, it introduces the problem of optimal dataset design, which is cast as a bilevel optimization problem. The optimal dataset design problem is motivated using theoretical insights on the approximation of functions by neural networks, relating the properties of a function describing a training dataset to the model capacity required to represent it.

3. It introduces a tractable algorithm for the generation of datasets that are amenable to learning, and empirical demonstration of marked improvements to the accuracy of trained models, as well as the ability to satisfy constraints at inference time.

4. Finally, it provides state-of-the-art accuracy results at vastly enhanced computational runtime on learning two challenging optimization problems: Job Shop Scheduling problems and Optimal Power Flow problems for energy networks.

To the best of the authors' knowledge, this work is the first to highlight the issue of learnability in the face of co-optimal or approximate solutions obtained to generate training data for learning to approximate hard CO problems. *The observations raised in this work may result in a broader impact as, in addition to approximating hard optimization problems, the optimal dataset generation strategy introduced in this paper may be useful to the line of work on integrating CO as differentiable layers for predictive and prescriptive analytics, as well as for physics constrained learning problems such as when approximating solutions to systems of partial differential equation.*

## 2 Related work

The integration of CO models into ML pipelines to meld prediction and decision models has recently seen the introduction of several nonoverlapping approaches, surveyed by Kotary et al. [19]. The application of ML to boost performance in traditional search methods through branching rules and enhanced heuristics is broadly overviewed by Bengio et al. [5]. Motivated also by the need for fast approximations to combinatorial optimization problems, the training of surrogate models via both supervised and reinforcement learning is an active area for which a thorough review is provided by Vesselinova et al. [31]. Designing surrogate models with the purpose to approximate hard CO problem has been studied in a number of works, including [6, 12, 15]. An important aspect for the learned surrogate models is the prediction of solutions that satisfy the problem constraints. While this is a very difficult task in general, several methodologies have been devised. In particular, Lagrangian loss functions have been used for encouraging constraint satisfaction in several applications of deep learning, including fairness [30] and energy problems [15]. Other methods iteratively modify training labels to encourage satifaction of constraints during training [12].

This work focuses on an orthogonal direction with respect to the literature reviewed above. Rather than devising a new methodology for effectively producing a surrogate model that approximate some hard optimization problem, it studies the machine learning task from a data generation perspective. It shows that the co-optimality and symmetries of a hard CO problem may be viewed as a source of noise that obscures the relationships between training data and the target outputs and proposes an optimal data generation approach to mitigate these important issues.

## 3 Preliminaries

A constrained optimization (CO) problem poses the task of minimizing an *objective function* $f : \mathcal{Y} \times \mathcal{X} \to \mathbb{R}_+$ of one or more variables $y \in \mathcal{Y} \subseteq \mathbb{R}^n$, subject to the condition that a set of *constraints* $C_x$ are satisfied between the variables and where $x \in \mathcal{X} \subseteq \mathbb{R}^m$ denotes a vector of input data that specifies the problem instance:

$$O(x) = \underset{y}{\mathrm{argmin}} \, f(y, x) \quad \text{subject to:} \quad y \in C_x. \tag{1}$$

An assignment of values $y$ which satisfies $C_x$ is called a *feasible solution*; if, additionally $f(y, x) \leq f(w, x)$ for all feasible $w$, it is called an *optimal solution*.

A particularly common constraint set arising in practical problems takes the form $C = \{y \, : \, Ay \leq b\}$, where $A \in \mathbb{R}^{m \times n}$ and $b \in \mathbb{R}^m$. In this case, $C$ is a convex set. If the objective $f$ is an affine function, the problem is referred to as *linear program* (LP). If, in addition, some subset of a problem's variables are required to take integer values, it is called *mixed integer program* (MIP). While LPs with convex objectives belong to the class of convex problems, and can be solved efficiently with strong theoretical guarantees on the existence and uniqueness of solutions [7], the introduction of integral constraints ($y \in \mathbb{N}^n$) results in a much more difficult problem. The feasible set in MIP consists of distinct points in $y \in \mathbb{R}^n$, not only nonconvex but also disjoint, and the resulting problem is, in general, NP-Hard. Finally, nonlinear programs (NLPs) are optimization problems where some of the constraints or the objective function are nonlinear. Many NLPs are nonconvex and can not be efficiently solved [24].

The methodology introduced in this paper is illustrated on hard MIP and nonlinear program instances.

## 4 Problem setting and goals

This paper focuses on learning approximate solutions to problem (1) via supervised learning. The task considers datasets $\chi = \{(x^{(i)}, y^{(i)})\}_{i=1}^N$ consisting of $N$ data points with $x^{(i)} \in \mathcal{X}$ being a vector of input data, as defined in equation (1), and $y^{(i)} \in O(x^{(i)})$ being a solution of the optimization task. A desirable, but not always achievable, property is for the solutions $y^{(i)}$ to be optimal.

The goal is to learn a model $f_\theta : \mathcal{X} \to \mathcal{Y}$, where $\theta$ is a vector of real-valued parameters, and whose quality is measured in terms of a nonnegative, and assumed differentiable, *loss function* $\ell : \mathcal{Y} \times \mathcal{Y} \to \mathbb{R}_+$. The learning task minimizes the empirical risk function (ERM):

$$\min_\theta J(f_\theta; \chi) = \frac{1}{N} \sum_{i=1}^N \ell(f_\theta(x^{(i)}), y^{(i)}), \tag{2}$$

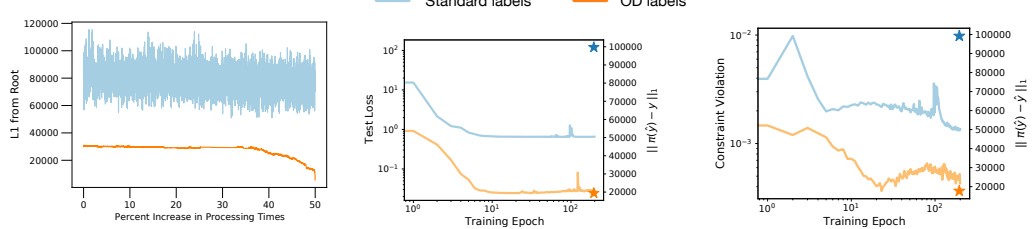

Figure 2: Dataset solution $y^{(i)}$ comparison: L1 Distance from a reference solution (left); Test loss (center); and Constraint violations (right).

with the desired goal that the predictions also satisfy the problem constraints: $f_\theta(x^{(i)}) \in C_x$.

While the above task is difficult to achieve due to the presence of constraints, the paper adopts a Lagrangian approach [14], which has been shown successful in learning constrained representations, along with projection operators, commonly applied in constrained optimization to ensure constraint satisfaction of an assignment. For a given point $\hat{y}$, e.g., representing the model prediction, a projection operator $\pi_C(\hat{y})$ finds the closest feasible point $y \in C$ to $\hat{y}$ under a $p$-norm:

$$\pi_C(\hat{y}) \stackrel{\text{def}}{=} \underset{y}{\mathrm{argmin}} \, \|y - \hat{y}\|_p \quad \text{subject to:} \quad y \in C.$$

The full description of the Lagrangian based approach and the projection method adopted is delegated to the Appendix A.

## 5 Challenges in learning hard combinatorial problems

One of the challenges arising in this area comes from the recognition that a problem instance may admit a variety of disparate optimal solutions for each input $x$. To illustrate this challenge, the paper uses a set of scheduling instances that differ only in the time required to process tasks on some machine. A standard approach to the generation of dataset in this context would consist in solving each instance independently using some SoTA optimization solver. However, this may create some significant issues that are illustrated in Figure 2 (more details on the problem are provided in Section 8). The blue curve in Figure 2 (left) illustrates the behavior of this natural approach. In the figure, the processing times in the instances increase from left to right and the blue curve represents the $L_1$-distance between the obtained solution to each instance (i.e., the start times of the tasks) and a reference optimal solution for some instance. The volatile curve shows that meaningful patterns can be lost, including the important relationship between an increase in processing times and the resulting solutions. Figure 2 (center) shows that, while the solution patterns induced by the target labels appear volatile, the ERM problem appears well behaved, in the face of minimizing the test loss. However, when training loss converges, accuracy (measured as the distance between the projection of the prediction $\pi_C(\hat{y})$ and the real label $y$) remains poor in models trained on such data (blue star). Figure 2 (right) shows the average magnitude of the constraints violation during training, corresponding to the two target solution sets of Figure 2 (left), along with a comparison of the objective of the projection operator applied to the prediction: $\|\pi_C(\hat{y}) - \hat{y}\|$. It is worth emphasizing that these issues are further exacerbated when time constraints prevent the solver from obtaining optimal solutions. Moreover, similar patterns can also be observed for the data generated while solving optimal power flow instances that exhibit symmetries.

Additionally, extensive observations collected on the motivating applications of the paper show that, even when the model complexity (i.e., the dimensionality of the model parameters $\theta$) is increased arbitrarily, the resulting learned models tend to have low-variance. This is illustrated in Figure 3, where the orange and blue curves depict, respectively, a function interpolating the training labels and the associated learned solutions.

Figure 3: Approximating highly volatile function results in low-variance models.

The goal of this paper is to construct datasets that are well-suited for learning the optimal (or near-optimal) solutions to optimization problems. The benefit of such an

approach is illustrated by the orange curves and stars in Figure 2, which were obtained using the data generation methodology proposed in this paper. They considered the same instances and obtained (different) optimal solutions, but exhibit much enhanced behavior on all metrics.

## 6 Theoretical justification of the data generation

Although the data generation strategy is necessarily heuristic, it relies on key theoretical insights on the nature of optimization problems and the representation capabilities on neural networks. This section reviews these insights.

First, observe that, as illustrated in the motivating Figure 3, the solution trajectory associated with the problem instances on various input parameters can often be naturally approximated by piecewise linear functions. This approximation is in fact exact for linear programs when the inputs capture incremental changes to the objective coefficients or the right-hand side of the constraints. Additionally, ReLU neural networks, used in this paper to approximate the optimization solutions, have the ability to capture piecewise linear functions [17]. While these models are thus compatible with the task of predicting the solutions of an optimization problem, the model capacity required to represent a target piecewise linear function exactly depends directly on the number of constituent pieces.

**Theorem 1** (Model Capacity [2]). *Let $f : \mathbb{R}^d \to \mathbb{R}$ be a piecewise linear function with $p$ pieces. If $f$ is represented by a ReLU network with depth $k + 1$, then it must have size at least $\frac{1}{2}kp^{\frac{1}{k}} - 1$. Conversely, any piecewise linear function $f$ that is represented by a ReLU network of depth $k + 1$ and size at most $s$, can have at most $\left(\frac{2s}{k}\right)^k$ pieces.*

The solution trajectories may be significantly different depending on how the data is generated. Hence, the more volatile the trajectory, the harder it will be to learn. Moreover, for a network of fixed size, the more volatile the trajectory, the larger the approximation error will be in general. The data generation proposed in this paper will aim at generating solution trajectories that are approximated by piecewise linear functions with fewer distinct pieces. The following theorem bounds the approximation error when using continuous piecewise linear functions: it connects the approximation errors of a piecewise linear function with the *total variation in its slopes*.

**Theorem 2.** *Suppose a piecewise linear function $f_{p'}$, with $p'$ pieces each of width $h_k$ for $k \in [p']$, is used to approximate a piecewise linear $f_p$ with $p$ pieces, where $p' \leq p$. Then the approximation error*

$$\|f_p - f_{p'}\|_1 \leq \frac{1}{2}h_{\max}^2 \sum_{1 \leq k \leq p} |L_{k+1} - L_k|,$$

*holds where $L_k$ is the slope of $f_p$ on piece $k$ and $h_{\max}$ is the maximum width of all pieces.*

*Proof.* Firstly, the proof proceeds with considering the special case in which $f_p$ coincides in slope and value with $f_{p'}$ at some point, and that each piece of $f_{p'}$ overlaps with at most 2 distinct pieces of $f_p$. This is always possible when $p' \geq \frac{p}{2}$. Call $I_k$ the interval on which $f_{p'}$ is defined by its $k^{th}$ piece. If $I_k$ overlaps with only one piece of $f_p$, then for $x \in I_k$,

$$|f_p(x) - f_{p'}(x)| = 0 \tag{3}$$

If $I_k$ overlaps with pieces $k$ and $k + 1$ of $f_p$, then for $x \in I_k$,

$$|f_p(x) - f_{p'}(x)| \leq h_k|L_{k+1} - L_k| \tag{4}$$

Each of the above follows from the assumption that $f_p$ and $f_{p'}$ are equal in their slope and value at some point within $I_k$. From this it follows that on $I_k$,

$$\|f_p - f_{p'}\|_1 = \int_{I_k} |f_p - f_{p'}| \leq \frac{1}{2} \sum_{1 \leq i \leq p} h_k^2|L_{k+1} - L_k| \leq \frac{1}{2}h_{\max}^2 \sum_{1 \leq k \leq p} |L_{k+1} - L_k|, \tag{5}$$

so that on the entire domain of $f_p$ and $f_{p'}$,

$$\|f_p - f_{p'}\|_1 \leq \frac{1}{2}h_{\max}^2 \sum_{1 \leq k \leq p} |L_{k+1} - L_k|. \tag{6}$$

Since removing the initial simplifying assumptions tightens this upper bound, the result holds. □

The final observation that justifies the proposed approach is the fact that optimization problems often satisfy a local Lipschitz condition, i.e., if the inputs of two instances are close, then they admit solutions that are close as well, i.e., there exist $\mathring{\boldsymbol{y}}^{(i)} \in O(\boldsymbol{x}^{(i)})$ and $\mathring{\boldsymbol{y}}^{(j)} \in O(\boldsymbol{x}^{(j)})$, where

$$\| \mathring{\boldsymbol{y}}^{(i)} - \mathring{\boldsymbol{y}}^{(j)} \| \le C \| \boldsymbol{x}^{(i)} - \boldsymbol{x}^{(j)} \|, \tag{7}$$

for some $C \ge 0$ and $\| \boldsymbol{x}^{(i)} - \boldsymbol{x}^{(j)} \| \le \epsilon$, where $\epsilon$ is a small value. This is obviously true in linear programming when the inputs vary in the objective coefficients or the right-hand side of the constraints, but it also holds locally for many other types of optimization problems. That observation suggests that, when this local Lipschitz condition holds, it may be possible to generate solution trajectories that are well-behaved and can be approximated effectively. Note that Lipschitz functions can be nicely approximated by neural networks as the following result indicates.

**Theorem 3** (Approximation [9]). *If $f : [0,1]^n \to \mathbb{R}$ is $L$-Lipschitz continuous, then for every $\epsilon > 0$, there exists some single-layer neural network $\rho$ of size $N$ such that $\|f - \rho\|_\infty < \epsilon$, where $N = \binom{n + \frac{3L}{\epsilon}}{n}$.*

The result above illustrates that the model capacity required to approximate a given function depends to a non-negligible extent on the Lipschitz constant value of the underlying function.

Note that the results in this section are bounds on the ability of neural networks to represent generic functions. In practice, these bounds themselves rarely guarantee the training of good approximators, as the ability to minimize the empirical risk problem in practice is often another significant source of error. In light of these results however, it is to be expected that datasets which exhibit less variance and have small Lipschitz constants[1] will be better suited to learning good function approximations. The following section presents a method for dataset generation motivated by these considerations.

## 7 Optimal CO training data design

Given a set of input data $\{\boldsymbol{x}^{(i)}\}_{i=1}^N$, the goal is to construct the associated pairs $\boldsymbol{y}^{(i)}$ for each $i \in [N]$, that solve the following problem

$$\min_{\theta, \boldsymbol{y}^{(i)}} \frac{1}{N} \sum_{i=1}^N \ell(f_\theta(\boldsymbol{x}^{(i)}), \boldsymbol{y}^{(i)}) \tag{8a}$$

$$\text{subject to : } \boldsymbol{y}^{(i)} \in \operatorname*{argmin}_{\boldsymbol{y} \in C_{\boldsymbol{x}^{(i)}}} f(\boldsymbol{y}, \boldsymbol{x}^{(i)}). \tag{8b}$$

One often equips the data point set $\{\boldsymbol{x}^{(i)}\}_{i=1}^N$ with an ordering relation $\preceq$ such that $\boldsymbol{x} \preceq \boldsymbol{x}' \Rightarrow \|\boldsymbol{x}\|_p \le \|\boldsymbol{x}'\|_p$ for some $p$-norm. For example, in the scheduling domain, the data points $\boldsymbol{x}$ represent task start times and the training data are often generated by "slowing down" some machine, which simulates some unexpected ill-functioning component in the scheduling pipeline. In the energy domain, $\boldsymbol{x}$ represent the load demands and the training data are generated by increasing or decreasing these demands, simulating the different power load requests during daily operations in a power network. For simplicity, this paper assumes the existence of such a useful ordering over the entire set of instances in a learning task.

From the space of co-optimal solutions $\boldsymbol{y}^{(i)}$ to each problem instance $\boldsymbol{x}^{(i)}$, the goal is to generate solutions which coincide, to the extent possible, with a target function of low total variation and Lipschitz factor, as well as a low number of constituent linear pieces in the case of discrete optimization. While it may not be possible to produce a target set that simultaneously optimizes each of these metrics, they are confluent and can be improved simultaneously. Natural heuristics are available which reduce these metrics substantially when compared with naive approaches.

One heuristic aimed at satisfying the aforementioned properties reduces to the problem of determining a solution set $\{\boldsymbol{y}^{(i)}\}_{i=1}^N$ for the inputs $\{\boldsymbol{x}^{(i)}\}_{i=1}^N$ of problem (1) that minimizes their total variation:

$$\text{minimize } TV\left(\{\boldsymbol{y}^{(i)}\}_{i=1}^N\right) = \frac{1}{2} \sum_{i=1}^{N-1} \|\boldsymbol{y}^{(i+1)} - \boldsymbol{y}^{(i)}\|_p \tag{9a}$$

$$\text{subject to : } \boldsymbol{y}^{(i)} = \operatorname*{argmin}_{\boldsymbol{y} \in C_{\boldsymbol{x}^{(i)}}} f(\boldsymbol{y}, \boldsymbol{x}^{(i)}). \tag{9b}$$

---

[1]The notation here is used to denote its discrete equivalent, as indicated in Equation (7).

In practice, this bi-level minimization cannot be achieved, due partially to its prohibitive size. It is possible, however, to minimize the individual terms of (9a), each subject to the result of the previous, by solving individual instances sequentially. Algorithm 1 ensures that solutions to subsequent instances have minimal distance with respect to the chosen $p$-norm (the experiments of Section 8 use $p = 1$). This method approximates a set of solutions with minimal total variation, while ensuring that the maximum magnitude of change between subsequent instances is also small. When the data represent the result of a discrete optimization, this coincides naturally with a representative function

---

**Algorithm 1:** *Opt. Data Generation*

$\quad$ **input :** $\{x^{(i)}\}_{i=1}^{N}$: Input data

1 $\quad y^{(N)} \leftarrow \overset{*}{y}^{(N)} \in \tilde{O}(x^{(N)})$

2 $\quad$ **for** $i = N - 1$ *down to* $1$ **do**

3 $\qquad y^{(i)} \leftarrow \overset{*}{y}^{(i)} \in \tilde{O}(x^{(i)})$

4 $\qquad y^{(i)} \in \begin{cases} \text{argmin}_{y} & \|y - y^{(i+1)}\|_p \\ \text{subject to:} & y \in C_{x^{(i)}} \\ & f(y) \le f(\overset{*}{y}^{(i)}) \end{cases}$

5 $\quad$ **return** $\chi = \left\{ \left( x^{(i)}, y^{(i)} \right) \right\}_{i=1}^{N}$

---

which requires less pieces, with less extreme changes in slope. The method starts by solving a target instance, e.g., the last in the given ordering $\preceq$ (line 1). Therein, $\tilde{O}$ denotes the solution set of a (possibly approximated) minimizer for problem (1). In the case of the job shop scheduling, for example, $\tilde{O}$ represents a local optimizer with a suitable time-limit. The process then generates the next dataset instance $y^{(i)}$ in the ordering $\preceq$ by solving the optimization problem given in line 3. The problem finds a solution to problem $x^{(i)}$ that is close to adjacent solution $y^{(i+1)}$ while preserving *optimality*, i.e., the objective of the sought $y$ is constrained to be at most that of $\overset{*}{y}^{(i)} \in \tilde{O}(x^{(i)})$.

When the difficulty of the underlying optimization instances makes the sequential solving of Algorithm 1 impractical, structural properties of the CO problem may be exploited to increase efficiency, i.e., by the use of warm-starts or solution-guided search. In the Job Shop Scheduling case study, $\overset{*}{y}^{(i+1)}$ is feasible to the subsequent problem and may be used to warm-start its solution, carrying forward solution progress between iterations of Algorithm 1. Therefore, a byproduct of this data-generation approach is that the optimization problem in line 3 can be well-approximated within a short timeout. When such exploits are not available, the secondary optimization of line 4 may be applied over independently pre-solved instances in an analogous way.

In addition to providing enhanced efficacy for learning, this method of generating target instances is generally preferable from a modeling point of view. When predicted solutions to related decision problems are close together, the resulting small changes are often more practical and actionable, and thus highly preferred in practice. For example, a small change in power demands should result in an updated optimal power network configuration which is easy to achieve given its previous state.

## 8 Application to case studies

The concepts introduced above are applied in this section to two representative case studies, *Job Shop Scheduling* (JSS) and *Optimal Power Flow* (OPF). Both are of interest in the optimization and machine learning communities as practical problems which must be routinely solved, but are difficult to approximate under stringent time constraints. The JSS problem represents the class of combinatorial problems, while the OPF problem is continuous but nonlinear and non convex. Both lack solution methods with strong guarantees on the rate of convergence, and the quality of solutions that can be obtained. In the studies described below, a deep neural ReLU network equipped with a Lagrangian loss function (described in details in Appendix A) is used to predict the problem solutions that are approximately feasible and close to optimal. Efficient projection operators are subsequently applied to ensure feasibility of the final output (See Appendix B and C).

**Job shop scheduling**

Job Shop Scheduling (JSS) assumes a set of $J$ jobs, each consisting of a list of $M$ tasks to be completed in a specified order. Each task has a fixed processing time and is assigned to one of $M$ machines, so that each job assigns one task to each machine. The objective is to find a schedule with minimal *makespan*, or time taken to process all tasks. The *no-overlap* condition requires that for any two tasks assigned to the same machine, one must be complete before the other begins. See the problem specification in Appendix B. The objective of the learning task is to predict the start times of all tasks given a JSS problem specification (task duration, machine assignments).

**Data Generation Algorithms** The experiments examine the proposed models on a variety of problems from the JSPLIB library [28]. The ground truth data are constructed as follows: different

| Instance | Size | Prediction Error | | Constraint Violation | | Optimality Gap (%) | | Time SoTA Eq. (s) | |
|---|---|---|---|---|---|---|---|---|---|
| | $J \times M$ | Standard | OD | Standard | OD | Standard | OD | Standard | OD |
| ta25 | 20×20 | 193.9 | **23.4** | 180.0 | **45.5** | 10.3 | **4.0** | 24 | **550** |
| yn02 | 20×20 | 153.2 | **38.9** | 124.9 | **70.3** | 9.1 | **4.5** | 27 | **45** |
| swv03 | 20×10 | 309.4 | **12.4** | 206.9 | **31.6** | 18.0 | **2.2** | 15 | **65** |
| swv07 | 20×15 | 330.4 | **19.9** | 280.1 | **67.2** | 17.0 | **3.0** | 15 | **60** |
| swv11 | 50×10 | 1090.0 | **51.2** | 906.4 | **151.7** | 28.5 | **4.5** | 13 | **100** |

Table 1: Standard vs OD training data: prediction errors, constraint violations, and optimality gap (the smaller the better), Time SoTA Eq. (the larger the better). Best results are highlight in bold.

input data $x^{(i)}$ are generated by simulating a machine slowdown, i.e., by altering the time required to process the tasks on that machine by a constant amount which depends on the instance $i$. Each training dataset associated with a JSS benchmark is composed of a total of 5000 instances. Increasing the processing time of selected tasks may also change the difficulty of the scheduling. The method of sequential solving outlined in Section 7 is particularly well-suited to this context. Individual problem instances can be ordered relative to the amount of extension applied to those processing times, so that when $d_{jt}^{(i)}$ represents the time required to process task $t$ of job $j$ in instance $i$, $d_{jt}^{(i)} \leq d_{jt}^{(i+1)}$ $\forall j, t$. In this case, any solution to instance $d_{(i+1)}$ is feasible to instance $d_i$ (tasks in a feasible schedule cannot overlap when their processing times are reduced, and start times are held constant). As such, the method can be made efficient by passing the solution between subsequent instances as a warm-start.

The analysis compares two datasets: One consisting of target solutions generated independently with a solving time limit of 1800 seconds using the state-of-the-art IBM CP Optimizer constraint programming software (denoted as Standard), and one whose targets are generated according to algorithm 1, called the Optimal Design dataset (denoted as OD).

Figure 4 presents a comparison of the total variation resulting from the two datasets. Note that the OD datasets have total variation which is orders of magnitude lower than their Standard counterparts. Recall that a small total variation is perceived as a notion of well-behaveness from the perspective of function approximation. Additionally, it is noted that the total computation time required to generate the OD dataset is at least an order of magnitude smaller than that required to generate the standard dataset (13.2h vs. 280h).

| Instance | Size | Total Variation ($\times 10^6$) | |
|---|---|---|---|
| | $J \times M$ | Standard Data | OD Data |
| ta25 | 20×20 | 67.8 | **0.194** |
| yn02 | 20×20 | 55.0 | **0.483** |
| swv03 | 20×10 | 109.4 | **0.424** |
| swv07 | 20×15 | 351.2 | **0.100** |
| swv11 | 50×10 | 352.0 | **1.376** |

Figure 4: Standard vs OD training data: Total Variation.

**Prediction Errors and Constraint Violations** Table 1 reports the prediction errors as $L_1$-distance between the (feasible) predicted variables, i.e., the projections $\pi(\hat{y})$ and their original ground-truth quantities ($y$), the average constraint violation degrees, expressed as the $L_1$-distance between the predictions and their projections, and the optimality gap, which is the relative difference in makespan (or, equivalently objective values) between the predicted (feasible) schedules and target schedules. All these metrics are averaged over all perturbed instances of the dataset and expressed in percentage. In the case of the former two metrics, values are reported as a percentage of the average task duration per individual instance. Notice that for all metrics the methods trained using the OD datasets result in drastic improvements (i.e., one order of magnitude) with respect to the baseline method. Additionally, Table 1 (last column) reports the runtime required by CP-Optimizer to find a value with the same makespan as the one reported by the projected predictions (projection times are also included). The values are to be read as the larger the better, and present a remarkable improvement over the baseline method. It is also noted that the worst average time required to obtain a feasible solution from the predictions is 0.02 seconds. Additional experiments, reported in Appendix D also show that the observations are robust over a wide range of hyper-parameters adopted to train the learning models.

*The results show that the OD data generation can drastically improve predictions qualities while reducing the effort required by a projection step to satisfy the problem constraints.*

**AC Optimal Power Flow**

*Optimal Power Flow (OPF)* is the problem of finding the best generator dispatch to meet the demands in a power network. The OPF is solved frequently in transmission systems around the world and is

| Instance | Size | Prediction Error | | Constraint Violation | | Optimality Gap (%) | |
|---|---|---|---|---|---|---|---|
| | No. buses | Standard | OD | Standard | OD | Standard | OD |
| Pegase-89 | 89 | 198.3 | **5.02** | 1.21 | **1.01** | 20.1 | **7.0** |
| IEEE-118 | 118 | 172.5 | **13.31** | 2.12 | **1.98** | 32.5 | **9.6** |
| IEEE-300 | 300 | 194.2 | **24.12** | 2.63 | **2.04** | 44.2 | **15.2** |

Table 2: Standard vs OD training data: prediction errors, constraint violations, and optimality gap.

increasingly difficult due to intermittent renewable energy sources. The problem is required to satisfy the AC power flow equations, that are non-convex and nonlinear, and are a core building block in many power system applications. The objective function captures the cost of the generator dispatch, and the Constraint set describes the power flow operational constraints, enforcing generator output, line flow limits, Kirchhoff's Current Law and Ohm's Law for a given load demand. The OPF receives its input from unit-commitment algorithms that specify which generators will be committed to deliver energy and reserves during the 24 hours of the next day. Because many generators are similar in nature (e.g., wind farms or solar farms connected to the same bus), the problem may have a large number of symmetries. If a bus has two symmetric generators with enough generator capacities, the unit commitment optimization may decide to choose one of the symmetric generators or to commit both and balance the generation between both of them. The objective of the learning task is to predict the generator setpoints (power and voltage) for all buses given the problem inputs (load demands).

**Data Generation Algorithms** The experiments compare this commitment strategy and its effect on learning on Pegase-89, which is a coarse aggregation of the French system and IEEE-118 and IEEE-300, from the NESTA library [10]. All base instances are solved using the Julia package PowerModels.jl [11] with the nonlinear solver IPOPT [33]. Additional data is reported in Appendix C. A number of renewable generators are duplicated at each node to disaggregate the generation capabilities. The test cases vary the load data by scaling the (input) loads from 0.8 to 1.0 times their nominal values. Instances with higher load pattern are typically infeasible. The unit-commitment strategy sketched above can select any of the symmetric generators at a given bus (Standard data). The optimal objective values for a given load are the same, but the optimal solutions vary substantially. Note that, when the unit-commitment algorithm commits generators by removing symmetries (OD data), the solutions for are typically close to each other when the loads are close. As a result, they naturally correspond to the generation procedure advocated in this paper.

**Prediction Errors and Constraint Violations** As shown in Table 2, the OD approach to data generation results in predictions that are closer to their optimal target solutions (error expressed in MegaWatt (MW)), reduce the constraint violations (expressed as $L_1$-distance between the predictions and their projections), and improve the optimality gap, which is the relative difference in objectives between the predicted (feasible) solutions and the target ones.

## 9 Limitations and Conclusions

This paper was motivated by engineering applications where problem instances share an underlying "infrastructure" (e.g., the power grid in optimal power flow or the manufacturing floor in job shop scheduling) which is stable and does not evolve too rapidly. These problems are pervasive in applications ranging from supply chains and logistics, to electricity grids, and manufacturing, to name a few. Note that the approach to data generation presented in this paper is not intended for direct application to every task of learning to solve constrained optimization. In general, data generation approaches should be tailored to exploit the properties of their respective optimization problems and experimental settings, but the algorithms and insights demonstrated above may be exploited in a variety of settings.

While the proposed methodology is limited to machine learning tasks in which optimization problem instances admit useful orderings with respect to their parameters, this property may hold only locally over a particular distribution of problems. For example, in timetabling applications, as in the design of employees shifts, a desired condition may be for shifts to be diverse for different but similar inputs. The proposed methodology may, in fact, induce a learner to predict similar solutions across similar input data. A possible solution to this problem may be that of generating various *trajectories* of solutions, learn from them with independent models, and then randomize the model selection to

generate a prediction. These more complex settings may require composition or extension of the solutions presented in this paper, and generalizations of the approach are an avenue for future work.

In addition to the inherent difficulty of learning feasible CO solutions, a practical challenge is represented by the data generation itself. Generating training datasets for supervised learning tasks requires solving many instances of hard CO problems, which can be time consuming and imposes a toll on energy usage and CO2 emissions. In this respect, an advantage of the proposed method, is its ability to use warm-starts to generate instances incrementally, resulting in enhanced efficiency even for hard optimization problems that require substantial solving time when treated independently.

Aside these limitations, the observations raised in this work may be significant in several areas: In addition to approximating hard optimization problems, the optimal dataset generation strategy introduced in this paper may be useful to the line of work on integrating CO and machine learning for predictive and prescriptive analytics, as well as for physics constrained learning problems, two areas with significant economic and societal impacts.

## Acknowledgments and Disclosure of Funding

This research is partially supported by NSF grant 2007164. Its views and conclusions are those of the authors only.

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
