## A    Lagrangian Dual-based approach

In both case studies presented below, a constrained deep learning approach is used which encourages the satisfaction of constraints within predicted solutions by accounting for the violation of constraints in a *Lagrangian* loss function

$$f_{\boldsymbol{\lambda}}(y) = f(y) + \sum_{i=1}^{m} \lambda_i \max(0, g_i(y)), \tag{10}$$

where $f$ is a standard loss function (i.e., *mean squared error*), $\lambda_i$ are *Lagrange multipliers* and $g_i$ represent the constraints of the optimization problem under the generic representation

$$\mathcal{P} = \underset{y}{\operatorname{argmin}}\, h(y) \quad \text{subject to} \quad g_i(y) \leq 0 \quad (\forall i \in [m]). \tag{11}$$

Training a neural network to minimize the Lagrangian loss for some value of $\lambda$ is anologous to computing a Lagrangian Relaxation:

$$LR_{\boldsymbol{\lambda}} = \underset{y}{\operatorname{argmin}}\, f_{\boldsymbol{\lambda}}(y), \tag{12}$$

and the *Lagrangian Dual* problem maximizes the relaxation over all possible $\lambda$:

$$LD = \underset{\boldsymbol{\lambda} \geq 0}{\operatorname{argmax}}\, f(LR_{\boldsymbol{\lambda}}). \tag{13}$$

The Lagrangian deep learning model is trained by alternately carrying out gradient descent for each value of $\lambda$, and updating the $\lambda_i$ based on the resulting magnitudes of constraint violation in its predicted solutions.

## B    Job Shop Scheduling

The Job Shop Scheduling (JSS) problem can be viewed as an integer optimization program with linear objective function and linear, disjunctive constraints. For JSS problems with $J$ jobs and $T$ machines, a particular instance is fully determined by the processing times $d_t^j$, along with machine assignments $\sigma_t^j$, and its solution consists of the resulting optimal task start times $s_j^t$. The full problem specification is shown below in the system (14). The constraints (14c) enforce precedence between tasks that must be scheduled in the specified order within their respective job. Constraints (14d) ensure that no two tasks overlap in time when assigned to the same machine.

### B.1    Problem specification

$$\mathcal{P}(\boldsymbol{d}) = \underset{\boldsymbol{s}}{\operatorname{argmin}}\, u \tag{14a}$$

$$\text{subject to: } u \geq s_T^j \qquad\qquad\qquad \forall j \in [J] \tag{14b}$$

$$s_{t+1}^j \geq s_t^j + d_t^j \qquad\qquad\qquad \forall j \in [J-1], \forall t \in [T] \tag{14c}$$

$$s_t^j \geq s_{t'}^{j'} + d_{t'}^{j'} \ \lor \ s_{t'}^{j'} \geq s_t^j + d_t^j \qquad \forall j, j' \in [J], t, t' \in [T] \text{ with } \sigma_t^j = \sigma_{t'}^{j'} \tag{14d}$$

$$s_t^j \in \mathbb{N} \qquad\qquad\qquad \forall j \in [J], t \in [T] \tag{14e}$$

Given a predicted, possibly infeasible schedule $\hat{s}$, the degree of violation in each constraint must be measured in order to update the multipliers of the Lagrangian loss function. The violation of task-precedence constraints (14c) and no-overlap constraint (14d) are calculated as in (15a) and (15b), respectively. Note that the violation of the disjunctive no-overlap condition between two tasks is measured as the amount of time at which both tasks are scheduled simultaneously on some machine.

$$\nu_{10b}\left(\hat{s}_t^j, d_t^j\right) = \max\left(0, \hat{s}_t^j + d_t^j - \hat{s}_{t+1}^j\right) \tag{15a}$$

$$v_{10c}\left(\hat{s}_t^j, d_t^j, \hat{s}_{t'}^{j'}, d_{t'}^{j'}\right) = \min\left(v_{10c}^L\left(\hat{s}_t^j, d_t^j, \hat{s}_{t'}^{j'}, d_{t'}^{j'}\right), v_{10c}^R\left(\hat{s}_t^j, d_t^j, \hat{s}_{t'}^{j'}, d_{t'}^{j'}\right)\right), \tag{15b}$$

where

$$v_{10c}^L\left(\hat{s}_t^j, d_t^j, \hat{s}_{t'}^{j'}, d_{t'}^{j'}\right) = \max\left(0, \hat{s}_t^j + d_j^t - \hat{s}_{t'}^{j'}\right)$$
$$v_{10c}^R\left(\hat{s}_t^j, d_t^j, \hat{s}_{t'}^{j'}, d_{t'}^{j'}\right) = \max\left(0, \hat{s}_{t'}^{j'} + d_{j'}^{t'} - \hat{s}_t^j\right).$$

The Lagrangian-based deep learning model does not necessarily produce feasible schedules directly. An additional operation is required for the construction of feasible solutions, given the direct neural network outputs representing schedules. The model presented below is used to construct solutions that are integral, and feasible to the original problem constraints. Integrality follows from the total unimodularity of constraints (16a, 16b), which converts the no-overlap condition of the problem (14) into addition task-precedence constraints following the order of predicted start times $\hat{s}$, denoted $\preceq_{\hat{s}}$. By minimizing the makespan as in (14), this procedure ensures optimality of the resulting schedules subject to the imposed ordering.

$$\Pi(s) = \quad \text{argmin}_s \ u$$
$$\text{subject to: } (14b), (14c)$$
$$s_t^j \geq s_{t'}^{j'} + d_{t'}^{j'} \qquad \forall j, j' \in [J], t, t' \in [T] \text{ s.t. } (j, t) \preceq_{\hat{s}} (j', t') \tag{16a}$$
$$s_t^j \geq 0 \qquad \forall j \in [J], t \in [T] \tag{16b}$$

## B.2 Dataset Details

The experimental setting, as defined by the training and test data, simulates a situation in which some component of a manufacturing system 'slows down', causing processing times to extend on all tasks assigned to a particular machine. Each experimental dataset is generated beginning with a root problem instance taken from the JSPLIB benchmark library for JSS instances. Further instances are generated by increasing processing times on one machine, uniformly over 5000 new instances, to a maximum of 50 percent increase over the initial values. To accommodate these incremental perturbations in problem data while keeping all values integral, a large multiplicative scaling factor is applied to all processing times of the root instance. Targets for the supervised learning are generated by solving the individual instances according to the methodology proposed in Section 7. A baseline set of solutions is generated for comparison, by solving individual instances in parallel with a time limit per instance of 1800 seconds.

The results presented in Section 8 are taken from the best-performing models, with respect to optimality of the predicted solutions following application of the model (16), among the results of a hyperparameter search. The model training follows the selection of parameters presented in Table 3.

| Parameter | Value | Parameter | Value |
|---|---|---|---|
| **Epochs** | 500 | **Batch Size** | 16 |
| **Learning rate** | $[1.25e^{-4}, 2e^{-3}]$ | **Batch Normalization** | False |
| **Dual learning rate** | $[1e^{-3}, 5e^{-2}]$ | **Gradient Clipping** | False |
| **Hidden layers** | 2 | **Activation Function** | ReLU |

Table 3: JSS: Training Parameters

## B.3 Network Architecture

The neural network architecture used to learn solutions to the JSS problem takes into account the structure of its constraints, organizing input data by individual job, and machine of the associated tasks. When $\mathcal{I}_k^{(j)}$ and $\mathcal{I}_k^{(m)}$ represent the input array indices corresponding to job $k$ and machine $k$, the associated subarrays $d[\mathcal{I}_k^{(j)}]$ and $d[\mathcal{I}_k^{(m)}]$ are each passed from the input array to a series of respective *Job* and *Machine layers*. The resulting arrays, one for every job and machine, are concatenated to form a single array and passed to further *Shared Layers*. Each shared layer has size $2JT$ in the case

**Model 1** $O_{\text{OPF}}$: AC Optimal Power Flow

---

$$\text{variables:} \quad S_i^g, V_i \ \ \forall i \in N, \ \ S_{ij}^f \ \ \forall (i,j) \in E \cup E^R$$

$$\text{minimize:} \quad O(\boldsymbol{S^d}) = \sum_{i \in N} c_{2i}(\Re(S_i^g))^2 + c_{1i}\Re(S_i^g) + c_{0i} \tag{17}$$

$$\text{subject to:} \quad \angle V_i = 0, \ \ i \in N \tag{18}$$

$$v_i^l \le |V_i| \le v_i^u \ \ \forall i \in N \tag{19}$$

$$\theta_{ij}^l \le \angle(V_i V_j^*) \le \theta_{ij}^u \ \ \forall (i,j) \in E \tag{20}$$

$$S_i^{gl} \le S_i^g \le S_i^{gu} \ \ \forall i \in N \tag{21}$$

$$|S_{ij}^f| \le s_{ij}^{fu} \ \ \forall (i,j) \in E \cup E^R \tag{22}$$

$$S_i^g - S_i^d = \textstyle\sum_{(i,j) \in E \cup E^R} S_{ij}^f \ \ \forall i \in N \tag{23}$$

$$S_{ij}^f = Y_{ij}^* |V_i|^2 - Y_{ij}^* V_i V_j^* \ \ \forall (i,j) \in E \cup E^R \tag{24}$$

---

of $J$ jobs and $T$ machines, and a final layer maps the output to an array of size $JM$, equal to the total number of tasks. This architecture improves accuracy significantly in practice, when compared with fully connected networks of comparable size.

## C  AC Optimal Power Flow

### C.1  Problem specification

*Optimal Power Flow (OPF)* is the problem of finding the best generator dispatch to meet the demands in a power network, while satisfying challenging transmission constraints such as the nonlinear nonconvex AC power flow equations and also operational limits such as voltage and generation bounds. Finding good OPF predictions are important, as a 5% reduction in generation costs could save billions of dollars (USD) per year [8]. In addition, the OPF problem is a fundamental building bock of many applications, including security-constrained OPFs [21]), optimal transmission switching [16], capacitor placement [4], and expansion planning [23].

Typically, generation schedules are updated in intervals of 5 minutes [29], possibly using a solution to the OPF solved in the previous step as a starting point. In recent years, the integration of renewable energy in sub-transmission and distribution systems has introduced significant stochasticity in front and behind the meter, making load profiles much harder to predict and introducing significant variations in load and generation. This uncertainty forces system operators to adjust the generators setpoints with increasing frequency in order to serve the power demand while ensuring stable network operations. However, the resolution frequency to solve OPFs is limited by their computational complexity. To address this issue, system operators typically solve OPF approximations such as the linear DC model (DC-OPF). While these approximations are more efficient computationally, their solution may be sub-optimal and induce substantial economical losses, or they may fail to satisfy the physical and engineering constraints.

Similar issues also arise in expansion planning and other configuration problems, where plans are evaluated by solving a massive number of multi-year Monte-Carlo simulations at 15-minute intervals [26, 13]. Additionally, the stochasticity introduced by renewable energy sources further increases the number of scenarios to consider. Therefore, modern approaches recur to the linear DC-OPF approximation and focus only on the scenarios considered most pertinent [26] at the expense of the fidelity of the simulations.

A power network $\boldsymbol{N}$ can be represented as a graph $(N, E)$, where the nodes in $N$ represent buses and the edges in $E$ represent lines. The edges in $E$ are directed and $E^R$ is used to denote those arcs in $E$ but in reverse direction. The AC power flow equations are based on complex quantities for current $I$, voltage $V$, admittance $Y$, and power $S$, and these equations are a core building block in many power system applications. Model 1 shows the AC OPF formulation, with variables/quantities shown in the complex domain. Superscripts $u$ and $l$ are used to indicate upper and lower bounds for variables. The objective function $O(\boldsymbol{S^g})$ captures the cost of the generator dispatch, with $\boldsymbol{S^g}$ denoting the vector of generator dispatch values ($S_i^g \mid i \in N$). Constraint (18) sets the reference angle to zero for the slack

| Instance | Size | | Total Variation | |
|---|---|---|---|---|
| | $\|N\|$ | $\|E\|$ | Standard Data | OD Data |
| **30_ieee** | 30 | 82 | 2.56570 | **0.00118** |
| **57_ieee** | 57 | 160 | 11.5160 | **0.00509** |
| **89_pegase** | 89 | 420 | 20.9309 | **0.02538** |
| **118_ieee** | 118 | 372 | 40.2253 | **0.01102** |
| **300_ieee** | 300 | 822 | 213.075 | **0.13527** |

Table 4: Standard vs OD training data: Total Variation.

bus $i \in N$ to eliminate numerical symmetries. Constraints (19) and (20) capture the voltage and phase angle difference bounds. Constraints (21) and (22) enforce the generator output and line flow limits. Finally, Constraints (23) capture Kirchhoff's Current Law and Constraints (24) capture Ohm's Law.

The Lagrangian-based deep learning model is based on the model reported in [15].

## C.2 Dataset Details

Table 4 describes the power network benchmarks used, including the number of buses $|\mathcal{N}|$, and transmission lines/transformers $|\mathcal{E}|$. Additionally it presents a comparison of the total variation resulting from the two datasets. Note that the OD datasets have total variation which is orders of magnitude lower than their Standard counterparts.

## C.3 Network Architecture

The neural network architecture used to learn solutions to the OPF problem is a fully connected ReLU network composed of an input layer of size proportional to the number of loads in the power network. The architecture has 5 hidden layers, each of size double the number of loads in the power network, and a final layer of size proportional to the number of generators in the network. The details of the learning models are reported in Table 5.

| Parameter | Value | Parameter | Value |
|---|---|---|---|
| **Epochs** | 20000 | **Batch Size** | 16 |
| **Learning rate** | $[1e^{-5}, 1e^{-4}]$ | **Batch Normalization** | True |
| **Dual learning rate** | $1e^{-4}$ | **Gradient Clipping** | True |
| **Hidden layers** | 5 | **Activation Function** | LeakyReLU |

Table 5: OPF: Training Parameters

# D Additional Results

Table 6 compares prediction errors and constraint violations for the OD and Standard approach to data generation for the Optimal Power Flow problems. As expressed in the main paper, the results show that the models trained on the OD datset present predictions that are closer to their optimal target solutions (error expressed in MegaWatt (MW)), reduce the constraint violations (expressed as $L_1$-distance between the predictions and their projections), and improve the optimality gap, which is the relative difference in objectives between the predicted (feasible) solutions and the target ones. Note that the results are improved from what reported in the main paper as a result of a larger hyper-parameter search with smaller learning rates and longer training times. The trends remain analogous to what is observed in the paper: The OD dataset induces dramatic improvements in both accuracy and constraint violation metrics.

| Instance | Size | Prediction Error | | Constraint Violation | | Optimality Gap (%) | |
|---|---|---|---|---|---|---|---|
| | No. buses | Standard | OD | Standard | OD | Standard | OD |
| IEEE-30 | 30 | 22.31 | **0.11** | 0.063 | **0.00004** | 6.28 | **0.76** |
| IEEE-57 | 57 | 83.61 | **0.58** | 0.139 | **0.0002** | 1.04 | **0.66** |
| Pegase-89 | 89 | 89.17 | **2.78** | 1.353 | **0.003** | 20.1 | **0.83** |
| IEEE-118 | 118 | 36.55 | **0.54** | 1.330 | **0.002** | 3.80 | **0.36** |
| IEEE-300 | 300 | 157.3 | **2.27** | 1.891 | **0.009** | 22.9 | **0.12** |

Table 6: OPF – Standard vs OD training data: prediction errors, constraint violations, and optimality gap.