# OpenReview forum: " Learning Hard Optimization Problems: A Data Generation Perspective"
_NeurIPS.cc/2021/Conference — NeurIPS 2021 Poster_

### Official Review · Reviewer_tnZz · 2021-07-12

**Rating:** 8
**Confidence:** 3

**Summary:**

Training predictive models for solutions of NP-hard constrained optimization problems is a very challenging task, since symmetric solutions are possible, solvers can exhibit strong non-deterministic behavior and solutions vary strongly although the instances are fairly similar. The authors of the paper at hand propose a new way of generating training data for supervised predictive models and show that their models outperform previous sota training-data-generation strategies on nonlinear nonconvex and hard-combinatorial problem instances under the challenges listed before.

**Limitations And Societal Impact:**

Only briefly addressed, but I think it is fine.

**Main Review:**

### Novelty

To the best of my knowledge, it’s one of the few papers -- maybe the first thorough paper -- that addresses the problem of training data generation in this context. The authors convincingly demonstrate potential challenges for this task and provide a theoretical analysis and a practical approach.

### Significance / Impact

In the last few years, the application of ML to constrained optimization has become a more and more important field of research and the authors clearly contribute to this. I believe that this paper both pushes the state of the art for the task clearly and opens up ample future research directions.

### Soundness

The theory, the proposed approach and the empirical analysis are sound and I don’t have found any major issues. In addition, the authors clearly state the assumption of their approach and convincingly discuss the limitations in the last section.
(Only the argument regarding energy consumption and CO2 seems a bit artificial, although still correct.)

### Clarity

The paper is very well written and easy to follow.

### Minor Comments

* “a optimal solution” → “an optimal solution”
* Figure 3 overlaps a bit with the text “symmetries”

### Questions for Rebuttal

1. How do the results in Table 1 and 2 compare to a real constraint solver?
2. Why is it better to maximize the time SOTA in Table 1?
3. Would this approach also be applicable to NP-hard decision problems such as SAT?


**Time Spent Reviewing:**

4

---

> ### Author Response · Authors · 2021-08-09
> **Response to Reviewer tnZz**
>
> Thank you for the constructive feedback. We are happy the paper was well received. We address your comments as follows and will be happy to discuss and clarify any further doubt you may have.
>
>
> **Question 1: Tables 1 and 2.**
>
> Both results on JSP and OPF are compared against state of the art industrial solvers (IBM CP-Optimizer for JSPs and COIN-OR IPOPT for OPFs).
> A modern constraint solver will eventually (given enough solving time) surpass the quality of solutions produced by ML models trained on both the Standard and OD datasets. We find that the results based on standard datasets are not competitive and are surpassed within a short time by real solvers, which partially motivates this work. To surpass results coming from OD-trained models on scheduling, for instance, CP-Optimizer can take anywhere from a few minutes to several hours. Comparing with the _SoTA runtime_ makes it possible to compare the proposed approach to the performance of highly optimized industrial solvers. The paper is motivated by contexts in which the same problem is solved repeatedly for different outputs, often in operational settings. This is the case in manufacturing, supply chains, logistics. and energy optimization where the OPF is solved every five minutes,
>
> **Question 2: About the SoTA time.**
>
> The _Time SoTA Eq_ column in Table 1 refers to the time required by CP-Optimizer to match the performance reported by the ML solution. Longer SoTA times, thus, correspond to predicted solutions of higher quality.
>
> **Question 3: Applicability to SAT.**
>
> In principle, even in the absence of an objective function as in SAT, there may still be many feasible solutions (symmetries) per problem instance, of which only one is chosen by the solver. So a similar challenge exists here.
> We believe that our results would apply to SAT encodings of feasibility problems that need to be solved repeatedly for classes of inputs. For instance, it could be very useful for timetabling problems that are encoded as SAT. However, this setting presents an additional challenge in that the SAT formulation encodes the input as well and the definition of similar instances is more complex in this case. This represents an interesting challenge and we thank the reviewer for asking this question.
>
> We hope this work may enable the study of this and similar questions, paving the way to further enhanced ML methods for solving difficult optimization problems.

---

> > ### Comment · Reviewer_tnZz · 2021-08-16
> > **Thanks**
> >
> > Thanks for the reply.
> > No further questions from my side.

---

### Official Review · Reviewer_vJwj · 2021-07-15

**Rating:** 7
**Confidence:** 4

**Summary:**

This paper addresses a different aspect of learning-based problem solvers.

Typical complex optimization problems are hard to find optimal solutions, and therefore learning datasets for a specific optimization problem class are volatile in general. Following this, the paper develops a method for optimal training data design, which is an optimization problem on the total variation of solutions. With the developed methods, two case studies (JSS and OPF) are demonstrated to show the effectiveness of their proposed methods.

**Limitations And Societal Impact:**

The paper covers some problem examples, and the paper and appendix explain the idea adopted. However, I guess that it is not trivial to follow the proposed approach for other problem classes.

**Main Review:**

Personally, I enjoyed reading this paper and am interested in this kind of approach.

- Pros
  - The targeting problem is novel and the author discusses a sound approach to tackle the problem.
  - The idea of using piecewise linear functions to represent the volatility is an interesting and effective approach for the problem illustrated in Fig.3.
  - Compared with the standard method to generate data sets, the proposed method could outperform such a baseline method.
- Cons
  - Explanations in challenges of the paper (Sec.5) are hard to follow. Many ambiguous terms (L1 from Root, Percent Increase in Processing Times, Test Loss, Constraint Violation, OD labels) are used, although the idea behind this part could be important to follow the problem set.
  - Some notations (see below comments) are confusing to clearly explain the idea.

### Questions and comments

- In Fig.1, three groups ($y^{(1)}_i, y^{(2)}_i, y^{(3)}_i$ for $i = 1, 2, 3, 4$) are illustrated, but the subscript $i$ is not explained explicitly. I conjecture that this corresponds to multiple solutions obtained from each instance $x^{(i)}$. What do the colors mean? (objective values?)
- In Eq. (1), $\mathcal{O}(x)$ is defined as the argmin of $f(y, x)$. When I first read this part, I assume that $\mathcal{O}(x)$ is a kind of value (not a set) which is a solution obtained after some methods (e.g., tie-braking, symmetry-braking, etc.). However, as in Sec.4, $y^{(i)}$ is in $\mathcal{O}(x)$ and $\mathcal{O}(x)$ is now a set of optimal solutions. I suggest noting explicitly $\mathcal{O}(x)$ is a (multi) set of optimal (or possibly sub-optimal) solutions to improve the readability.
    - This clarification could be important also for the projection operator $\pi_C$ as well because it is also defined with the argmin statement (i.e., operator finds 'the closest feasible point' = element).
    - This causes another problem in Eq. (9a) and (9b) because the $p$-norm among sets (if $y^{(i)}$ is a set) is not defined.
- In Algorithm 1, the loop $i$ starts with $N-1$ but $f(y) \leq f(y^\star (i))$ does not make sense as that of $i = N-1$ is not defined a priori. Does it mean $i+1$ or others?
- The paper assumes that similar instances generate similar optimal results and the bound of $i$-th instance can be inferred from $i+1$-th instance. As noted, this kind of relationship could hold for LPs. In Sec. 6, the authors said a kind of local Lipschitz condition holds also for many other types of optimization problems. Please clarify the scope of this word; 'many other types' to show the generalization ability of the proposed method.
- The standard approach assumes independence among data. However, for example in combinatorial optimization literature, re-optimization or life-long optimization is a branch of the research domain. Therefore, I suspect that the used standard approach is a bit week (a naive baseline). Do you have any other choice?
- Some minor comments
    - What does 'OD labels' mean in Figure 2? (Optimal Data generation?)
    - A missing sentence behind Figure 3 (symmetries?).
    - Figure 4 should be captioned as a table.

**Time Spent Reviewing:**

4

---

> ### Author Response · Authors · 2021-08-09
> **Response to Reviewer vJwj**
>
> Thank you for the constructive feedback. We are happy the paper was well received. We address your comments as follows and will be happy to discuss and clarify any further doubt you may have.
>
>
> - **Figure 1**: Same colors indicate possible solutions to the same inputs. Thus, each $y_i^{(1)}$ $(i \in \\{1,2,3\\})$ are colored identically as they are all solutions to input $x^{(1)}$.
>
> - **Equation 1**: We agree and will define ${\cal O}(x)$ as a multi-set of solutions earlier in the text. We also agree with your subsequent comments and the abuse of notation will be noted.
>
> - **Algorithm 1**: The symbol $y^{\*(i)}$ implies the solving of problem instance $i$ to optimize the objective. This is where $ y^{\*(N-1)}$ comes from. $y^{\*(N)}$ comes from Line 1 of Algorithm 1. To clarify, the bound on $f(y^{\*(N)})$ serves the purpose of maintaining optimality while performing the secondary optimization. We will make these aspects more explicit in Algorithm 1 (See also replies to  "_Algirhtm 3_" and "_L267_" to reviewer `EWPX`).
>
> - **Similar instances**: Thanks for this feedback. The bound coming from previously solved instances is not necessary for the method to work, and is not represented in Algorithm 1, but is used in our case studies as the relationship holds in those problems. We will clarify this language.
> Note also that the problem considered in this work, and generally many engineering problems that need to be solved repeatedly, do typically admit solutions that are close to each other. This includes manufacturing, energy optimization, supply chains, and logistics problems. However, due to symmetries, the reported solutions to close inputs may be arbitrary diverse. *This is a core aspect observed and addressed by this paper*.
>
> - **Baseline**: We have tried with both warm-started solutions and using heuristics. For the OPF problem, we noticed that warm-starting does not affect the reported results. In the JSP domain, we have also tried the adoption of heuristic methods that naturally make similar decisions in similar problems. We first mention for clarity that CP-optimizer is used in favor of fast heuristics to generate training data as they can produce optimal or near-optimal solutions if given enough time. As a result, our learning framework learns a better baseline than one given by heuristics. When we compared the makespan reported by several heuristics methods (short processing times, most work remaining, and most operations remaining) against that of CP-optimizer at 30-min (same settings as in the paper), the least such relative makespan increase among all heuristics on any dataset is 25.4%, and the typical case is well over 50% and up 770%. *They thus degrade the quality of the learning process substantially*.
>
> **Minor comments**
> - Yes, OD means Optimal Data generation. We will expand.
> - That is correct. The sentence ends with the word `symmetries.` (its last letter `s.` was covered by the image). We hope it has not impacted the reading. It will be fixed.
> - We agree. Many thanks!

---

> > ### Comment · Reviewer_vJwj · 2021-08-29
> > **Update after reading the response**
> >
> > Thank you for your helpful response to clarify my understanding (e.g., notations and baselines in experiments).
> >
> > Particularly, I'm interested in the symmetry and similar instances issue as we discussed. After reading the response, I was convinced that the addressed point in the paper is an interesting and important problem for the community. Of course, some important questions (e.g., the applicability of this method to a bunch of optimization problems under assumptions) are needed and studied more, but I still feel that the proposed approach is new and interesting. So, I'd like to keep my score and push the discussion to accept the paper.

---

> > > ### Author Response · Authors · 2021-08-29
> > > **Re: Update after reading the response**
> > >
> > > Thank you!

---

### Official Review · Reviewer_EWPX · 2021-07-16

**Rating:** 7
**Confidence:** 4

**Summary:**

This paper proposes a method to optimize training data generation for supervised learning approaches to constrained optimization. First the authors show that the existence of various optimal solutions to the problem can make the learning more challenging. Then they Then they proceed with some theoretical insights that link this challenge to the representation capacity of ReLU neural networks. The paper introduces a method for training data generation for CO learning such that CO instances that are close (in terms of specifications) have solutions that are close too. Numerical validation is presented on a job shop scheduling problem (combinatorial optimization) and an Optimal Power Flow problem (non-linear, non-convex constrained pb).


**Limitations And Societal Impact:**

Yes

**Main Review:**

# Originality

The problem of data generation optimization in the context of CO learning seems original and the proposed bi-level optimization and then its approximation too. There is a related work which handles symmetry and diversity in optimal CO solutions that could be cited:

Li et al, Combinatorial Optimization with Graph Convolutional Networks and Guided Tree Search, NeurIPS 2018

# Quality
The paper is of high quality. The problem of data generation and its influence on the learning task is well motivated. I appreciated the intuition given by the figures, the theoretical justification and the numerical experiments.

My main concern is about the applicability of the proposed method in general.

* In Algo 1 L3, where does y^\star^(i) comes from? If it is a bound based y^\star^(i+1) then it should be more explicit. Otherwise that would imply that 2 optimization problems are solved at each iteration?
* In L267 “a bound on y^\star^(i) can be inferred from y^\star^(i+1)” do you mean a bound on the value f(y^\star^(i)) ?
* Besides, also L267: warm starting the solver of the new instance with the solution of the previous instance only works if this latter is feasible, which is not guaranteed in general.

* The fact that this bound can be obtained easily is essential for the method to work. I wonder how realistic is this assumption.
* In the job shop scheduling illustration, I understand that a training set is generated just by “slowing down“ one machine, then there is a way to order the instances such that the assumption holds.
* However this setting does not seem very realistic to me. I would guess that the job shop scheduling is solved repeatedly with different jobs. But if the jobs characteristics are say uniformly sampled in some intervals, it is not clear how to get such order.
* Another CO problem that is often used as benchmark for learning-based heuristics is the Vehicle Routing Problem (and its variants). Related instances may have the same customers (nodes) but with a different demand in each instance. For a fixed vehicle capacity, the solution of 1 instance is likely to be infeasible for another instance so it won’t provide a bound nor an initial solution to warm-start and accelerate the solver.

* Because the data generation is focused on minimizing the training error, I wonder if the method has an impact on the generalization ability of the learned model to a slightly different test set?
* Tables 1 and 2: are the reported prediction errors evaluated on the training or test set?

# Clarity

The paper is generally well-written. I have a few questions and suggestions below:

## Statements to clarify:

* L29-32: the sentence implies that complex CO problems in realistic settings cannot be solved by OR and AI methods. While I may agree that AI-based optimisation may not (yet) be used in such applications, OR methods are of course applied.

* L41-41: “learning good CO approximations in jointly training prediction and optimization models”. I don’t understand this
* L95: What is meant by “surrogate models” here? This terminology is not used in the cited survey [23]
* L198: what is meant by *small* piecewise linear functions?
* What is the input to the Optimal Power Flow problem? In L347 it is “which generators will be committed to deliver energy and reserves during the 24 hours of the next day” then in L354 it’s the “load demands”. I thought the load demands were the input of the unit-commitment problem. I am confused…
* Figure 2 left: “percent increase in Processing Times” Are the processing times here the objective to minimise or a parameter of the instance? The problem not being defined at this point, I think it would be easier to understand if generic terms such that “objective”, “instance parameter”, etc are used. Or the correspondence clearly stated.
* Figure 3: same remark for “Generator setpoint” and “Base load factor”
* L212 “if the inputs of two instances are close, then they admit solutions that are close as well” and then equation (7): as it is written here, this contradicts the fact the initial observation that instances that are close can have very different solutions. I guess the authors mean *there exist* 2 solutions y(i) and y(j) that satisfy equation (7)

## Typos/details:

* L37: intersection *of*
* L48: may not have a *unique* (?) optimal solution
* L124: in y \in R^n —> remove in or y
* L268, L377 (and others): hot-start —> I believe the term is warm-start
* L225: the training *of* good…
* L365: for —> form (or remove)
* L379: *is* its restriction
* L368: when compared the OD approach to *standard* (?) data generation results
* L368: ... data generation results in predictions that are closer —> not clear
* Equation 7: which norm is used?

# Significance

The idea of this paper is original and interesting. I have a concern about its applicability though. I would have loved to see it illustrated on a strong learning-based method for a CO problem and see that it indeed lead to a better *test* performance.

# Update after rebuttal

The authors have addressed my main concerns about the applicability and limitations of their approach.

Since they mean to update their manuscript accordingly, I am happy to revise my score from 6 to 7.




**Time Spent Reviewing:**

7 (main review) + 2 (rebuttal)

---

> ### Author Response · Authors · 2021-08-09
> **Response to Reviewer EWPX**
>
>
> Thank you for the constructive feedback. We are happy the paper was well received. We address your applicability comment next and then reply to the other questions/comments by Section, following the style of your review.
>
> ### Applicability
>
> The paper is motivated by engineering applications in which the same problem is solved repeatedly for different inputs, often in operational settings. This is the case in manufacturing, supply chains, logistics, and energy optimization where the OPF is solved every five minutes. These systems operate continuously and have mechanisms (sometimes called _recourses_) to handle contingencies.
> The energy grid, for instance, is planned to sustain what is called N-1 contingencies (i.e., the loss of a generator or the loss of a line). But even if a more dramatic event occurs, the energy grid also can shed load to maintain operations. In logistics (since you are mentioning vehicle routing), the models almost always assume what is called a _complete recourse_: there exists a mechanism to handle all realizations of the uncertainty. In the example mentioned, it is always assumed that a truck can return to the depot if it can no longer accommodate the demand of the customers. The truck can then resume the planned tour subsequently. This is what stochastic optimization models assume to handle uncertainty. An analogous assumption could be made if we were to apply the proposed approach to a vehicle routing problem.
> We thank the reviewer for this observation and will make the connection to recourse in the paper.
>
> **On the existence of a bound.**
>
> Note that the proposed method **does not** require solutions ${y^\star}^{(i+1)}$ to warm-starting the search for problem with input $x^{(i)}$. L268 mentions that warm-starting _may_ be used to speed up the solver. However, our implementation **does not** use this strategy, yet, the resolution for problem with input $x^{(i)}$ is extremely fast as driven by the solution ${y^\star}^{(i+1)}$.
>
> The _bound_ really just comes from the optimal solution of the original problem, so it is always available. The purpose of using it as a bound is to maintain optimality while solving the secondary problem. We use the notation $y*^{(i)}$ to imply that instance $i$ has been solved optimally to produce this value. We realize that this notation can be improved and will write explicitly the solving step into Algorithm 1 in the final version of the paper.
>
> Finally, we also note that the methodology does not require a total ordering. It simply requires a notion of distance between solutions, which is natural in the contexts studied in the paper.
> The algorithm proposed is an illustration that deals with many practical cases in which a total ordering exists, but it can be generalized to a broader context. For example, the algorithm may start with an arbitrary instance and then greedily select the closest instance (in the input space) at each step. The central concept is to reduce arbitrary variations over the space of target solutions.
> We will make this important point clearer to emphasize the generality of the approach. Many thanks for the valuable remark.
>
>
> ### Quality
>
> - **Algorithm 3 L3**: This is a second optimization problem for each iteration which retains optimality while trying to minimize disparity from the previously calculated solution. This secondary problem can be set with a lower timeout limit since it is not critical to solving it very close to optimality.
>
> - **L267**: When a bound between subsequently solved problems is available (as in the cases studied in the paper), we apply the bound as mentioned in L267, but this step is not necessary in theory and is not included in Algorithm 1. The bound of Algorithm 1 (L3) is to maintain optimality of the primary objective (i.e., makespans in scheduling or generator costs in OPFs) as mentioned above.
> We recognize our writing can be improved in L267 and will clarify it in our revised version.
>
> - **VRP**: The VRP mentioned is interesting and the team has expertise in deployments in such contexts. As mentioned above, these problems are often modeled to include a recourse that ensures feasibility. Once again we stress that this is not an artifact of the proposed solution. These problems are modeled this way in stochastic optimization.
>
> - **Complex CO problems**: We also want to note that the job-shop scheduling problems are extremely complex CO problems. The proposed framework produces solutions that are close to optimal within milliseconds and is the first to tackle the JSP with supervised learning, to the best of our knowledge, thanks to the proposed data generation approach. It paves the way to what we believe may be an interesting avenue for the use of deep learning to solve complex scheduling problems.
>
> - **Generalization**: Our experiments (Tables 1 and 2 and appendix) report results on the **test sets** (and thus use instances not observed during training). The results were also cross-validated, using 5-fold cross-validation, with no perceptible change on prediction errors.
>
>
> ### Clarity
>
> We agree that the sentences pointed out could be better explained. We will further clarify all the following in our final version. We will also address all the typos. Thank you!
>
> - **L29-32**: We agree and will be more specific about this comment. Note also that the paper focuses on the setting in which many repeated instances of the problem must be solved under very stringent time constraints. This scenario challenges even highly optimized optimization solvers, as demonstrated in our experiments.
>
> - **L41-41**: This area of research refers to the predict-and-optimize philosophy, where learning models, aided with external optimization solvers or with the use of implicit layers, are trained end-to-end to solve some data-driven decision problems.
>
> - **L95**: By "surrogate model" we mean a model which can be used to predict solutions to an optimization problem. The term has been used in previous work (e.g., [15]) but we will be more explicit.
>
> - **L198 "small"**: We mean piecewise linear function with small extreme changes in slope.
>
> - **OPF inputs**: The commitments are parameters to the OPF, and the loads are its inputs. Symmetries arise in the generator dispatch itself.
>
> - **Figure 2**: The processing times are a parameter of the instance; The objective is to minimize the makespan. We will also clarify what parameters and objectives are for the OPF earlier in the text.
>
> - **L212**: The problems considered in this work, and generally many engineering problems that need to be solved repeatedly, do typically admit solutions that are close to each other for inputs which also close to one another.
> What we noticed, however, is that even in the contexts where similar inputs should intuitively produce similar outputs, symmetries or suboptimality due to a solver time limit, may create variability in the solution space which complicates the learning. This is a core aspect observed and addressed by this paper. We however agree that the clarity of this sentence may be improved and we will correct it.
>
> Once again, we have appreciated your time and comments! We hope we have resolved all your concerns and will be happy to answer further questions or any doubt you may have.

---

> > ### Comment · Reviewer_EWPX · 2021-08-31
> > **Response to authors**
> >
> > I thank the authors for their thorough reply.
> >
> > I still have a couple of points that I would like to understand better.
> >
> > 1. The authors say in their reply “L268 mentions that warm-starting may be used to speed up the solver. However, our implementation does not use this strategy, yet, the resolution for problem with input x(i) is extremely fast as driven by the solution y*(i+1).”
> >
> > What do they mean exactly by “driven by the solution y*(i+1)”?
> >
> >
> >
> > 2. The authors say “We use the notation y^*(i) to imply that instance i has been solved optimally to produce this value.”
> >
> > This means that 2 optimization problems are solved for each instance i:
> > * Problem (1) is solved to get the value f(y*(i)) (possibly, as in the use-cases, using y(i+1) to warm-start the solver, but this can only be done if y(i+1) is feasible for instance i)
> > * The problem at line (3) is solved to compute another optimal solution y(i) that is as close as possible to y(i+1), Possibly this problem is solved approximately with a smaller solver timeout.
> >
> > Is this correct?
> >
> > This would mean that in general the proposed Optimal Design dataset generation may be more costly than the standard approach. However, at least in the JSS case study, it results in a significant speed-up (L322). For me this can only be explained by the efficient warm-starting — but the authors say that they do not use it currently (see my point 1.), so I am confused…
> >
> > This speedup is also mentioned in the conclusion as an advantage of the approach: L376: “an advantage of the proposed method, is its ability to use hot-starts to generate instances incrementally, resulting in enhanced efficiency even for hard optimization problems that require substantial solving time when treated independently.”
> >
> > But this is only true if the warm-starting can be used (i.e. given 2 instances that are "close", if the optimal solution of one instance is feasible for the other instance). Since this is not true in general, I find the above statement misleading.
> >
> > I would appreciate any clarifications about these points.

---

> > > ### Comment · Reviewer_EWPX · 2021-08-31
> > > **About Applicability and Recourse**
> > >
> > > In the context of the paper, assuming the uncertain parameters are the instance parameters x, complete recourse would mean that whatever the realization x, there exists a feasible solution y to problem (1). This sounds reasonable enough, although it might not be a necessary assumption, since the parameters x are chosen explicitly in the generation process.
> > >
> > > The questions in my review and my previous comment are around the assumption that an optimal solution y(i+1) of problem (1) for x=x(I+1) would be feasible for problem (1) with x=x(i). I can’t see the link with the recourse.
> > >
> > > I think that making the connection to the recourse might be confusing, especially since the framework of stochastic optimization is not explicitly used in the paper.

---

> > > > ### Author Response · Authors · 2021-09-01
> > > > **Re: Response to authors**
> > > >
> > > > Thank you for the corrections and feedback!
> > > >
> > > > **On the issue of warm-starts:**
> > > > We apologize for not being clearer and for the confusion it created. Let us go over this issue carefully. What we meant by “driven by the solution $y^*$$^{(i+1)}$"  is that, in some case studies (e.g., the jobshop) we communicate to the solver the solution $y^{*(i+1)}$ as a starting point (e.g., a MIP Start or a CP Starting Point) to the problems with input $x^{(i)}$. These starting points do not need to be feasible---*in fact, they are not always feasible in our test cases*---but they may be used by the solver to guide the search towards feasible solutions.
> > > >
> > > > These starting points are communicated to the CP solver (e.g., using the method `setStart()`) in the jobshop experiments.
> > > > Technically, this is different from a warm-start, in that the latter is assumed to be feasible, and this also contributed to the confusion in our response.
> > > > For instance, in the OPF problem, where we use an interior point method, $y^{*(i+1)}$ is not necessarily feasible for the problems with input $x^{(i)}$ and warm-starts are not used in our OPF experiments. It is in fact not useful, in general, to communicate these points to IPOPT (which ideally needs an interior point/warm-start). This is what we meant when we said that we are not using a warm-start, and again, we acknowledge we should have been clearer.
> > > >
> > > > Thus, your intuition that the JSP study uses "warm-starts" (with the caveat mentioned above) is correct.
> > > >
> > > > Your understanding of the two bulleted comments in your question 2 are also correct.
> > > >
> > > > **About “Applicability and Recourse”:**
> > > > We hope that our comments above also clarify your last question about applicability and recourse. We agree that we should not refer to stochastic optimization in the final version.
> > > >
> > > > We thank the reviewer for the feedback. It has helped us realize what and how the language used in our presentation should be improved.

---

> > > > > ### Comment · Reviewer_EWPX · 2021-09-01
> > > > > **Re: Re: Response to Authors**
> > > > >
> > > > > Thank you for the clarifications.
> > > > >
> > > > > I think that it would be important to report in the paper the time it takes to generate the data (standard versus proposed Optimal Design) for the Optimal Power Flow case study. That way, we will have an illustration of the “cost of the OD” in the two possible scenarios: when the solutions of previous problems can be exploited to speedup the process (Job Shop) and when they cannot (Opt Power Flow).
> > > > >
> > > > > The possible additional time is a drawback or limitation of the proposed approach. Spending more time to generate optimized training data might make sense in some applications (such as the OPF), as it can lead to a significant gain in performance. In some other applications, if the standard generation cost is already too high (and no speedup can be achieved by exploiting previous solutions) maybe it won’t be possible to use this OD strategy. Therefore I believe it’s important to mention this limitation explicitly.
> > > > >
> > > > > I still think the paper has a great and original contribution. Assuming the authors agree to update their manuscript to take into account our discussion, I will be happy to increase my score to 7.

---

> > > > > > ### Author Response · Authors · 2021-09-01
> > > > > > **Thank you!**
> > > > > >
> > > > > > We agree with the reviewer, and, in fact, we had independently planned to do so for the final version of the paper!
> > > > > > We will also add a discussion about the potential limitations this aspect entails and how we think they can be circumvented in some cases (e.g., using a bound derived from solving the previous instance in the ordering, when the problem structure allows it).
> > > > > >
> > > > > > We are genuinely glad for your feedback. We believe these updates will improve the paper presentation and outline interesting avenues for future research.
> > > > > > Many thanks again!

---

### Official Review · Reviewer_WzHp · 2021-07-16

**Rating:** 7
**Confidence:** 4

**Summary:**

This paper proposes a method to generate solutions to optimization problems that are more amenable to supervised learning tasks.

**Limitations And Societal Impact:**

Please see the main review

**Main Review:**

This paper proposes a method to generate solutions to optimization problems that are more amenable to supervised learning tasks. That is to say, when the parameters of a given problem gradually change, the solutions found also gradually change, such that the resultant data points are stable and learnable. This paper is interesting overall and the proposed method is simple yet effective.

1.	The key assumption is that optimization problems typically satisfy a local Lipschitz condition, i.e., if the inputs of two instances are close, then they admit solutions that are close as well. However, this may not be the case for graph problems, where a single local perturbation could result in significant change to the solution.

2.	Figure 3 covers the last part of line 165.

3.	The figures in the paper should be explained clearly. For example, in Figure 3 what is “Generator setpoint”? Since main part of Section 6 is motivated by Figure 3, it is hard to get the points of Section 6.

4.	Can you explain why in Figure 2 (center) the training loss of blue one is so high, compared to its test loss? It seems the training process can indeed improve the test performance even in case of Standard Labels. Does it mean that some patterns do exit in the training set, although they are not obvious to human but can be learned.


**Time Spent Reviewing:**

4h

---

> ### Author Response · Authors · 2021-08-09
> **Response to Reviewer WzHp**
>
> Thank you for the constructive feedback. We address your comments as follows and will be happy to discuss and clarify any further doubt you may have.
>
> **Question 1: Assumption.**
>
> The problems considered in this work, and generally many engineering problems that need to be solved repeatedly, do typically admit solutions that are close to each other for inputs which also close to one another.
> What we noticed, however, is that even in the contexts where similar inputs should intuitively produce similar outputs, symmetries or suboptimality due to a solver time limit, may create variability in the solution space which complicates the learning. Since feedforward neural networks learn piecewise linear functions, and the local Lipschitz factor of any learned function depends on the magnitude of the jumps made by the underlying step function (see Theorems 2 and 3), our proposal modifies the data generation process to attenuate the volatility of the problem, i.e., the jump magnitudes.
>
> For graph problems, it really depends on how the instances differ. For example, if the solutions to the graph problem represent a flow and the inputs differ by the demand in some nodes (e.g., some min capacity constraints on some edges), the proposed approach should work naturally. In fact, the optimal power flow is a sophisticated form of a graph problem. If the problem is a matching on graphs whose topologies change drastically for every instance, even after considering symmetries, approximating its solutions via machine learning models may not be a good choice. The work in this paper is motivated by engineering applications where instances typically have some similarity because the underlying “infrastructure” (e.g., the power grid or the manufacturing floor) is stable and does not evolve too rapidly. These problems are pervasive in applications ranging from supply chains and logistics, to electricity grids, and manufacturing, to name a few.
>
> **Question 2: Figure covering text.**
>
> We apologize! The sentence ends with the word `symmetries.` (its last letter `s.` was covered by the image). We hope it has not impacted the reading. It will be fixed.
>
> **Question 3: Figures explanation.**
>
> We take heed to your feedback about better explaining the figures and will improve their description. "Generator setpoint" is a term used by power systems engineers, with whom we collaborate, and it indicates the physical quantities required to operate a generator. These are the active power associated with the generator and the voltage magnitude associated with the bus in which the generator resides. It can be considered as a "value assignment" for that generator.
>
> **Question 4: Figure 2 (center).**
>
> Note that Figure 2 does not report the _training losses_.
> Figure 2 (center) compares the **test loss** of the standard label generation (blue curve) against the proposed OD label generation (orange curve).
> The test losses improve in that average errors are indeed minimized during the learning process, albeit not significantly. In fact, they still result in predictions that are inaccurate and induce large constraint violations (see Figure 2 right). Additionally while not reported, the training losses are comparable, in magnitude and trends, with their respective test losses.
>
> We hope we have resolved all your concerns and will be happy to answer further questions or any doubt you may have.

---

> > ### Comment · Reviewer_WzHp · 2021-08-26
> > **Thanks**
> >
> > I appreciate the clarifications from the authors. I have raised my score.
> >
> > One suggestion: I think the following motivation is important for readers to capture the application scenarios of the proposed method, which should be highlighted in Introduction.
> >
> > >this paper is motivated by engineering applications where instances typically have some similarity because the underlying “infrastructure” (e.g., the power grid or the manufacturing floor) is stable and does not evolve too rapidly. These problems are pervasive in applications ranging from supply chains and logistics, to electricity grids, and manufacturing, to name a few.
> >
> > Also, I want to mention that there have been some works [1,2] studying data generation in the filed of learning to optimize. Although these papers focus on the generalization ability of the trained solvers (which are different from this work), they could be discussed in the related work.
> >
> > [1] S Liu et al., Generative Adversarial Construction of Parallel Portfolios. TCYB 2020.
> >
> > [2] K Tang et al., Few-Shots Parallel Algorithm Portfolio Construction via Co-Evolution. TEVC 2021.

---

> > > ### Author Response · Authors · 2021-08-27
> > > **Re: Thanks**
> > >
> > > Thank you for your positive feedback and for your suggestion. We agree and will modify the Introduction to introduce this motivation sentence.
> > >
> > > We also appreciated the additional references. They will be discussed in the related work section in the final version.
> > >
> > > Again, many thanks!

---

### Decision · Program_Chairs · 2021-09-27

**Decision:**

Accept (Poster)

**Comment:**

This paper provides a methodology for generating datasets for  supervised learning for combinatorial optimization problems. This is an interesting perspective as it focuses on the importance of generating the right datasets for the supervised learning problem. The authors highlight several challenges that characterize  combinatorial optimization problems and provide some theoretical insights for the proposed methodology. In particular the authors propose an approach that formulates the problem of optimal dataset design as a bilevel  optimization problem and introduces an efficient algorithm for  dataset generation. They show the effectiveness of the approach  for job shop scheduling and optimal power flow problems. There was unanimous consensus on accepting the paper.